# A Combined Strategy of Improved Variable Selection and Ensemble Algorithm to Map the Growing Stem Volume of Planted Coniferous Forest

Xiaodong Xu [1,2,3], Hui Lin [1,2,3], Zhaohua Liu [1,2,3], Zilin Ye [1,2,3], Xinyu Li [1,2,3] and Jiangping Long [1,2,3,*]

[1]  Research Center of Forestry Remote Sensing & Information Engineering, Central South University of Forestry and Technology, Changsha 410004, China; xuxiaodong@csuft.edu.cn (X.X.); T19911090@csuft.edu.cn (H.L.); 20171100032@csuft.edu.cn (Z.L.); 20181100028@csuft.edu.cn (Z.Y.); lxy365@csuft.edu.cn (X.L.)

[2]  Key Laboratory of Forestry Remote Sensing Based Big Data & Ecological Security for Hunan Province, Changsha 410004, China

[3]  Key Laboratory of State Forestry Administration on Forest Resources Management and Monitoring in Southern Area, Changsha 410004, China

*   Correspondence: longjiangping@csuft.edu.cn; Tel.: +86-0731-8562-3848

**Abstract:** Remote sensing technology is becoming mainstream for mapping the growing stem volume (GSV) and overcoming the shortage of traditional labor-consumed approaches. Naturally, the GSV estimation accuracy utilizing remote sensing imagery is highly related to the variable selection methods and algorithms. Thus, to reduce the uncertainty caused by variables and models, this paper proposes a combined strategy involving improved variable selection with the collinearity test and the secondary ensemble algorithm to obtain the optimally combined variables and extract a reliable GSV from several base models. Our study extracted four types of alternative variables from the Sentinel-1A and Sentinel-2A image datasets, including vegetation indices, spectral reflectance variables, backscattering coefficients, and texture features. Then, an improved variable selection criterion with the collinearity test was developed and evaluated based on machine learning algorithms (classification and regression trees (CART), k-nearest neighbors (KNN), support vector regression (SVR), and artificial neural network (ANN)) considering the correlation between variables and GSV (with random forest (RF), distance correlation coefficient (DC), maximal information coefficient (MIC), and Pearson correlation coefficient (PCC) as evaluation metrics), and the collinearity among the variables. Additionally, we proposed a secondary ensemble with an improved weighted average approach (IWA) to estimate the reliable forest GSV using the first ensemble models constructed by Bagging and AdaBoost. The experimental results demonstrated that the proposed variable selection criterion efficiently obtained the optimal combined variable set without affecting the forest GSV mapping accuracy. Specifically, considering the first ensemble, the relative root mean square error (rRMSE) values ranged from 21.91% to 30.28% for Bagging and 23.33% to 31.49% for AdaBoost, respectively. After the secondary ensemble involving the IWA, the rRMSE values ranged from 18.89% to 21.34%. Furthermore, the variance of the GSV mapped by the secondary ensemble with various ranking methods was significantly reduced. The results prove that the proposed combined strategy has great potential to reduce the GSV mapping uncertainty imposed by current variable selection approaches and algorithms.

**Keywords:** growing stem volume; sentinel; variable selection; ensemble algorithm

## 1. Introduction

With the aggravation of global warming, resources and environmental sustainability have become the most concerning problem [1–3]. Given the decrease in natural forests, planted forests are regarded as the most critical ecological system on land, playing an indispensable role in reducing carbon dioxide concentration [4,5]. The forest growing stock

volume (GSV) is a crucial indicator of the quality of planted forests [6–8]. Typically, field-measured methods are considered the most accurate way to estimate GSV, but the process is time-consuming and labor-intensive [6,7]. In the last decade, remote sensing images have been applied in mapping forest GSV, with the GSV estimation accuracy being affected by ground measured forest parameters, remote sensing images, extracted variables, and estimated models. Compared with the last three factors, the ground measurement errors are pretty small and thus negligible when using precision measuring approaches. To reduce the uncertainty of mapping GSV, selecting the appropriate remote sensing images, variables, and models depending on the various forest and external environments is mandatory.

Currently, light detection and ranging (LiDAR), synthetic aperture radar (SAR), and optical remote sensing imagery are the three primary data sources for mapping forest GSV [6–8]. LiDAR data have a significant advantage in estimating the vertical forest structure, especially canopy height [9]. However, it is over costly for applying LiDAR on a large scale. SAR images relying on C-and L-band frequency have been proven to be more effective in mapping forest parameters because of their capability to penetrate forests [6]. Moreover, the optical satellite images with high spatial resolution contain more detailed spatial information and present great potential for mapping forest parameters [10,11]. Considering the cost factor, medium-resolution SAR and optical data with a high temporal resolution are more suitable for mapping forest GSV on a large scale [6,8].

Given the variety of variables available from multiple remote sensing data sources (vegetation indices, spectral reflectance variables, backscattering coefficients, and texture features), variable selection methods are the critical step to obtain the optimally combined variables for forest GSV estimation, with the accuracy of the results highly depending on the sensitivity of the selected variables [6,7,12]. To date, the variable selection methods can be broadly classified into three categories: filters, wrappers, and embedded [13]. The filters work independently on the learner regardless of its specifics. The selection of variables is based on some algorithms, such as RF, DC, MIC and PCC [7,14–16]. However, determining the thresholds of a single criterion is controversial [17]. In a wrapper approach to variable selection, the evaluation of a candidate variable subset is obtained based on its usefulness in training, and the whole process is performed during the training set [18,19]. The optimal variable subset is extracted by repeatedly training models, while the involved data processing is time-consuming [20]. Finally, the embedded method together with elimination form a learning system. The weight of each variable is obtained during the training set, and the variable selection is based on this weight. As in a decision tree, some variables are selected at each node, and this process is part of the algorithm that cannot be separated from it. However, the weight coefficients of the set of variables are highly correlated with the model used. Nevertheless, the weight coefficients and thresholds of the variable set are highly related to the employed models [21]. Hence, it is essential to obtain the optimal variable combination for mapping the forest GSV efficiently.

Commonly, estimating the GSV is highly related to the employed models [7,8,22]. In the past, some parametric models, such as the linear and non-linear models, were commonly employed to map forest parameters [23–25]. With the increase in remote sensing images and the number of extracted variables, non-parametric models, including CART, RF, K-NN, SVR, and ANN, were also widely used to map forest GSV [7,26,27]. These machine learning algorithms have significant advantages in solving complex, non-linear, and highly uncertain problems [28–31]. However, the accuracy and reliability of the estimated GSV are highly linked to the selected models. To overcome the shortcomings of a single model such as instability and overfitting, ensemble algorithm, e.g., RF [29], Bagging [32], and AdaBoost [33], have been widely used to improve the model's generalization ability [34,35]. Stable results can be obtained for the same base learners by changing the training times with Bagging or AdaBoost. On the other hand, combining the advantages of different types of base models has also been proposed, with the results usually being derived from base models utilizing the Voting or Stacking methods. Hence, the ensemble algorithm

presents a great capability to reduce the uncertainty caused by the employed base models and ultimately provide a reliable forest GSV.

Spurred by the deficiencies of current methods, this study proposed a combined strategy of improved variable selection and ensemble algorithm for mapping forest GSV. The suggested strategy can efficiently obtain the optimally combined variables and reduce the model uncertainty. Based on Sentinel-1A and Sentinel2-A data, we developed a criterion to connect machine learning models (CART, KNN, SVR, ANN) with standard methods for evaluating the variable importance (RF, DC, MIC, and PCC). After constructing the first ensemble models by Bagging and AdaBoost, the secondary ones involved an IWA capable of estimating a reliable forest GSV.

The remainder of this paper is as follows. Section 2 presents the materials and methods considered in this paper and the proposed variable selection criterion and IWA. Section 3 presents the experimental results, while Section 4 presents the discussion. Finally, Section 5 concludes this work.

## 2. Materials and Methods

### 2.1. Study Area

This work considered the Wangyedian experimental forest farm (total area of 25,958 ha), located in the southwest of Chifeng City, Inner Mongolia (118°09′–118°30′ E, 41°21′–41°39′ N), which as shown in Figure 1. The mid-temperate continental monsoon climate characterizes this region, and the mean annual sunshine duration is up to 2913.3 h. The topography of the forest farm is mainly hilly, with an altitude of 800 m~1890 m. The percentage of forest cover in the study area was about 93% by the end of 2016, with a total stock volume of 1.527 million m³. The area of the planted coniferous forest is about 49.78%, and the main planted tree species are larch *(Larix principis-rupprechtii* and *Larixolgensis)* and Chinese pine *(Pinus tabuliformis)*.

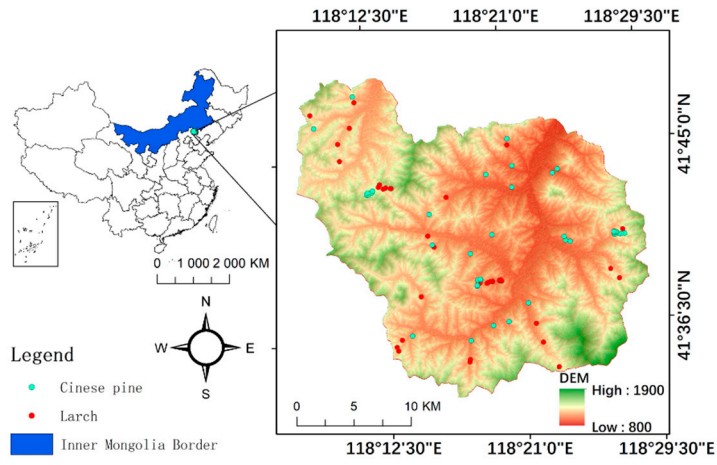

**Figure 1.** The location of the study area and the spatial distribution of ground samples.

### 2.2. Framework of This Research

We reduced the influence of the utilized variable selection method and model by suggesting a combined strategy that involved several variable selection criteria and ensemble algorithms in enhancing the GSV estimation. Figure 2 illustrates the combined strategy framework. Our study was divided into three parts: data preparation and variable extraction, variable selection, and constructing the secondary ensemble models.



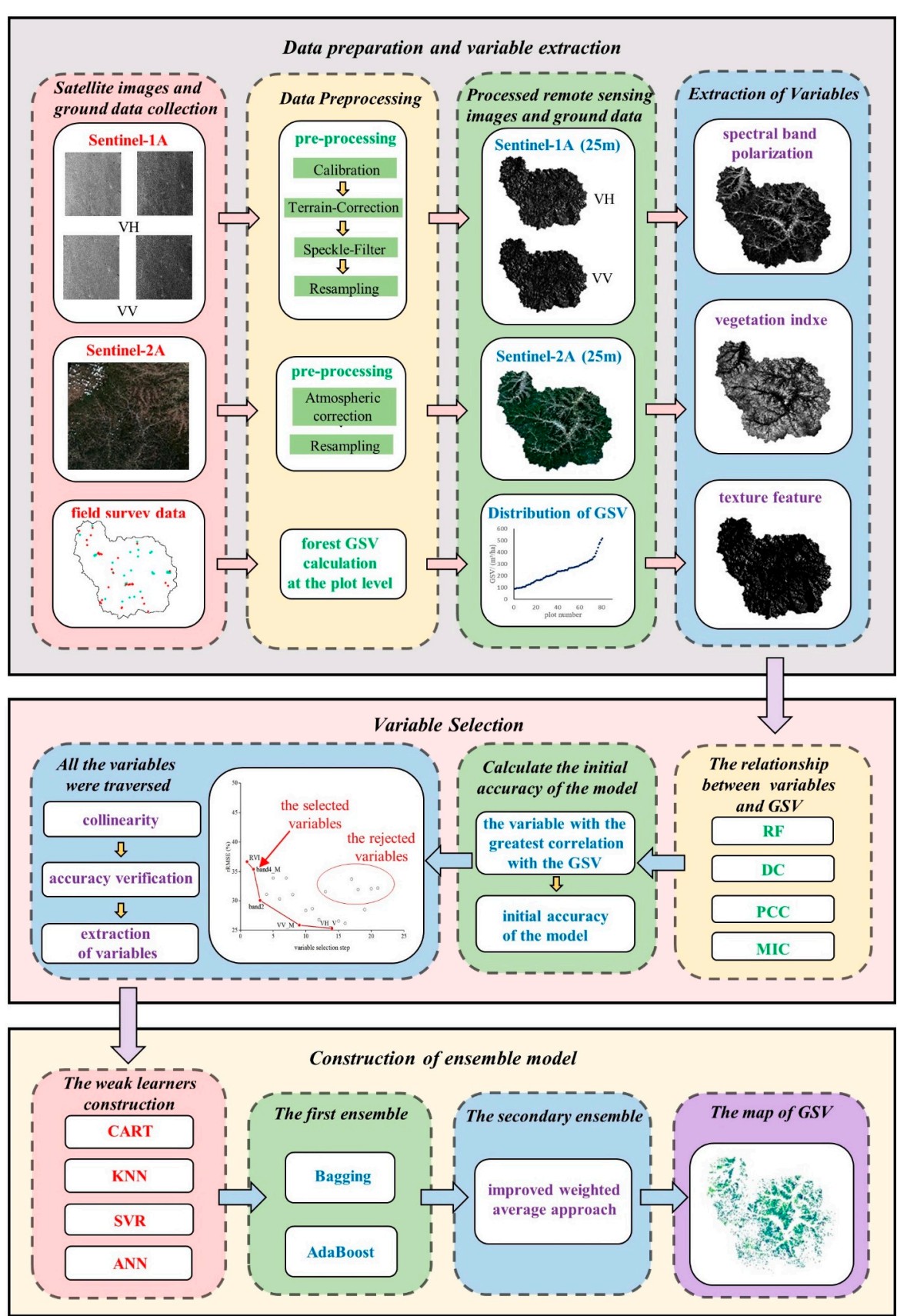

**Figure 2.** The framework of forest GSV estimation by a combined strategy involving improved variable selection and the secondary ensemble algorithm.

### 2.3. Ground Data Collection and Processing

Based on the age and spatial distribution of forest GSV, 81 ground samples were measured in September 2017 utilizing random stratification sampling in the study area (Figure 1). The sample size was 25 m × 25 m, with the position of the four corners and the central point measured through a Global Positioning System (GPS). The height and DBH of each tree were measured, and the volume of each tree was derived from the equations related to the measured height and DBH of the samples. The volumetric equations for the Wangyedian forest farm were defined as follows:

$$\text{Larch: V} = -0.001498 + 0.00007 \times D^2 + 0.000901 \times H + 0.000032 \times H \times D^2 \qquad (1)$$

$$\text{Chinese pine: V} = 0.013464 - 0.001967 \times D + 0.000089 \times D^2 + 0.000628 \times D \times H + 0.000032 \times H \times D^2 - 0.003173 \times H \quad (2)$$

where H, D, and V denote the measured height, DBH, and volume per tree. The GSV of the sample was the sum of all the tree timber volume within the sample. In our study, the GSV of the Larch samples varied between 86.17 m$^3$/ha and 405.56 m$^3$/ha, while the average GSV was 208.42 m$^3$/ha. The GSV of the Chinese pine samples varied between 91.97 m$^3$/ha and 355.61 m$^3$/ha, with the average GSV being 253.32 m$^3$/ha (Table 1).

**Table 1.** The GSV statistics of sample plots. (Unit: m$^3$/ha).

| Tree Species | Number of Plots | The Range of GSV | The Average GSV | STD |
|---|---|---|---|---|
| Larch | 38 | 86.17~405.56 | 208.38 | 81.84 |
| Chinese pine | 43 | 91.97~514.96 | 253.32 | 112.75 |

### 2.4. Remote Sensing Images and Pre-Processing

This study considered two cloud-free images of Sentinel-1A and Sentinel-2A acquired on September 19 and 22 September 2017, respectively, which are matched with the corresponding ground measurements. Table 2 illustrates the spectral band (Band2-Band8A) from Sentinel-2A and the polarizations (VH and VV) from Sentinel-1A used in this paper, matched for spatial resolution (25 m × 25 m). By employing the SNAP 8.0 software, the radiometric calibration and the Lee sigma filter (size of 7 × 7) were initially processed to reduce the speckle noise of the Sentinel-1A imagery. Then, the Range-Doppler terrain correction technology was used to correct the distance distortion caused by the scene terrain change and satellite sensor tilt. Regarding the Sentinel-2A imagery, the radiation correction, geometric correction, and atmospheric correction were applied to reduce the errors caused by the influence of interference factors.

**Table 2.** The information of the acquired remote sensing data.

| Sensors | Acquisition Date | Spectral Bands/Polarizations |
|---|---|---|
| Sentinel-1A (level-1GRD) | 19 September 2017 | VH, VV |
| Sentinel-2A (level-1C) | 22 September 2017 | Band2, Band3, Band4, Band5, Band6, Band7, Band8, Band8A |

### 2.5. Extraction of Variables

After pre-processing the images, four types of alternative variables, including vegetation indices, the values of spectral reflection, backscattering coefficients, and texture features, were extracted from the Sentinel-1A and Sentinel-2A imagery (Table 3). The spectral reflection values were directly derived from band 2 to band 8A for Sentinel-2A, and six commonly used vegetation indices were obtained through the mathematical operation of related bands, including Enhanced Vegetation Index (EVI) [36], Enhanced Vegetation Index-2 (EVI-2) [37], Normalized Difference Vegetation Index (NDVI) [38], Ratio Vegetation Index (RVI) [39], Spectral Vegetation Index (SVI) [40] and Soil Adjusted Vegetation Index

(SAVI) [41]. Moreover, backscattering coefficients of VH and VV polarization were extracted from Sentinel-1A, and the ratio of VH to VV was also calculated. Additionally, using the Gray Level Co-occurrence Matrix (GLCM), eight texture features (mean, variance, uniformity, contrast, dissimilarity, entropy, second moment, and correlation) were calculated with a size of $3 \times 3$ in the spectral reflection values, and backscattering coefficients.

**Table 3.** Variables extracted from Sentinel-1A and Sentinel-2A.

| Variable Type | Variable Name | Description of Variables | Sensors |
|---|---|---|---|
| Vegetation Index | Enhanced Vegetation Index (EVI) | $2.5 \times (Band8 - Band4)/(Band8 + 6\ Band4 - 7.5 \times Band2 + 1)$ | Sentinel-2A |
| | Enhanced Vegetation Index-2 (EVI-2) | $2.5 \times (Band8 - Band4)/(Band8 + 2.4 \times Band4 + 1)$ | Sentinel-2A |
| | Normalized Difference Vegetation Index (NDVI) | $(Band8 - Band4)/(Band8 + Band4)$ | Sentinel-2A |
| | Ratio Vegetation Index (RVI) | $Band8/Band4$ | Sentinel-2A |
| | Spectral Vegetation Index (SVI) | $Band4/Band8$ | Sentinel-2A |
| | Soil Adjusted Vegetation Index (SAVI) | $(1 + L) \times (Band8 - Band4)/(Band8 + Band4 + L)$ $L = 0.5$ in most conditions | Sentinel-2A |
| Spectral reflection | Spectral bands | Band2, Band3, Band4, Band5, Band6, Band7, Band8, Band8A | Sentinel-2A |
| Features of SAR | Backscattering coefficient | VH, VV, VH/VV | Sentinel-1A |
| Texture features | Mean, Variance, Contrast, Entropy, Homogeneity, Dissimilarity, Entropy, Second moment, Correlation | Gray Level Co-occurrence Matrix (GLCM) with size of $3 \times 3$ | Sentinel-1A Sentinel-2A |

*2.6. Proposed Variable Selection Criterion*

Typically, the number and types of the selected variables are highly dependent on the variable selection approaches. It is necessary to select an optimal variables combination for obtaining a reliable forest GSV and construct the models. However, the filter method selects variables by specific quantified criteria without considering the model's accuracy, such as RF, DC, MIC and PCC. On the other hand, the wrappers that focus on the model's accuracy will be time-consuming. For example, the forward selection method, which only considers accuracy, is denoted by FORW in this paper. This method iterates through all the variables and selects variables with the greatest gain until all the added variables fail to increase the model's accuracy. To overcome these drawbacks, the autocorrelation and interaction of independent variables were considered to reduce the operating time. Thus, we proposed an improved variable selection criterion involving a collinearity test that combined machine learning models with standard methods that evaluated the variable's importance. The proposed variable selection criterion involves the following steps:

Step 1: All alternative variables are ranked by selecting a single criterion (calculated by RF, DC, MIC, and PCC).

Step 2: The ranked first variable was initially selected as the most critical variable, and the initial $rRMSE_0$ was calculated by the employed model and the most critical variable.

Step 3: Then, the most critical variable was again selected among the remaining alternative variables and was tested for collinearity (VIF) with each selected variable. If VIF > 10, proceed to step 4. Otherwise, the variable is excluded, and step 3 is repeated for the remaining alternative variables.

Step 4: The selected variable that satisfies the collinearity test is added to the selected variable set, and the $rRMSE_1$ is calculated using the updated combined variables set. If $rRMSE_1 < rRMSE_0$, the variable is defined as valid. Otherwise, this variable is excluded.

Step 5: Repeat step 3 and step 4. The variables whose contribution decreased the rRMSE values are added to the combined variables set.

In the proposed criterion, the combined variables set were efficiently selected using the collinearity test. The capability of the proposed criterion regarding variable selection for estimating GSV depended on the single criterion for ranking and the algorithms. Therefore, four ranking criteria (RF, DC, MIC, and PCC) and four machine learning algorithms (CART, KNN, SVR, and ANN) were employed during processing. Additionally, we selected the FORW method as the reference method during the experiments.

### 2.7. Secondary Ensemble with Improved Weighted Average Approach

#### 2.7.1. Secondary Ensemble Algorithm

The estimated GSV accuracy is highly related to the employed models. To reduce this dependence, four machine learning models (CART, KNN, SVR, and ANN) were utilized as base learners, and then, we constructed the first ensemble models utilizing Bagging and AdaBoost. Therefore, we extracted four ensemble models using Bagging and four ensemble models using AdaBoost by the optimal combined variables sets.

After the first ensemble, the validated rRMSE values of the eight models were regarded as weight coefficients. The estimated GSV uncertainty imposed by the different models was reduced through the proposed secondary ensemble involving an IWA, defined as:

$$\hat{Y}_i = \sum_{i=1}^{n} \hat{y}_i \frac{(1 - rRMSE_i)}{\sum_{i=1}^{n}(1 - rRMSE_i)} \tag{3}$$

where $\hat{Y}_i$ and $\hat{y}_i$ are the estimated GSV from the secondary and first ensemble models, respectively, $rRMSE_i$ is the value of rRMSE from the first ensemble models, and n is the number of selected first ensemble models. Specifically, the selected first ensemble model that did not decrease the rRMSE values for forest GSV was neglected. The first ensembles were substituted into Equation (3) according to its validated rRMSE value from smallest to largest. All *n* values (1–8) are traversed to minimize the rRMSE value of the secondary ensemble model. The construction process of the secondary ensemble model is illustrated in Figure 3.

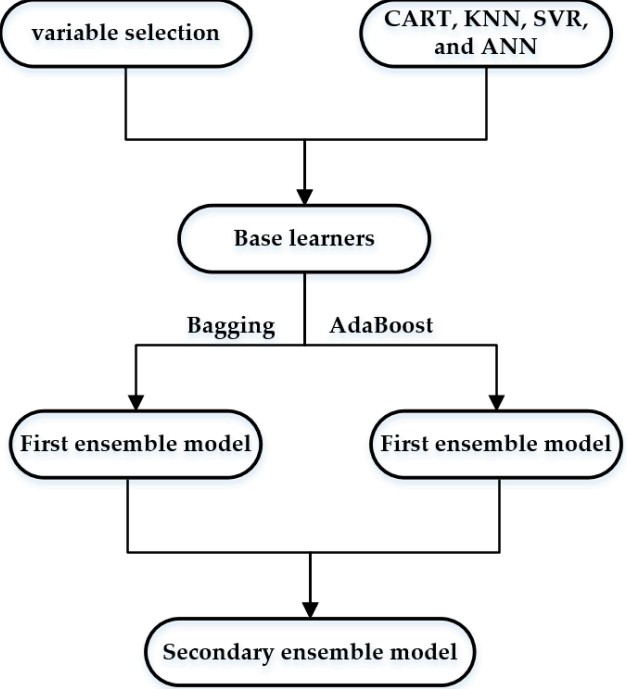

**Figure 3.** The flowchart of the secondary ensemble.

2.7.2. Accuracy Evaluation

To evaluate the estimated GSV of the various approaches, all ground measured samples were divided into three sets: training (36 samples), validation (18 samples), and testing (27 samples). We randomly repeated 100 times, selecting the training and validation sets to reduce the uncertainty for forest GSV estimation. Then, the rRMSE and determination coefficient ($R^2$) were calculated using the test set by averaging the results of each employed model. These indexes were calculated as follows:

$$\text{rRMSE} = \frac{\sqrt{\frac{\sum_{i=1}^n (y_i - \hat{y}_i)^2}{n}}}{\overline{y}} \tag{4}$$

$$R^2 = 1 - \frac{\sum_{i=1}^n (y_i - \hat{y}_i)^2}{\sum_{i=1}^n (y_i - \overline{y})^2} \tag{5}$$

## 3. Results

### 3.1. The Results of Variables Selection

In our study, 99 variables were extracted from Sentinel-1A and Sentinel-2A, and four standard methods of variable importance evaluation were employed, including RF, DC, MIC, and PCC, to rank all variables, such as vegetation indices, the spectral reflection values, backscattering coefficients, and texture features. The rRMSE values varied with the number of selected variables but determining the number of variables for estimating forest GSV was controversial. Figure 4 illustrates that for the single criterion case, the accuracy of the estimated GSV varied with the number of combined variables.

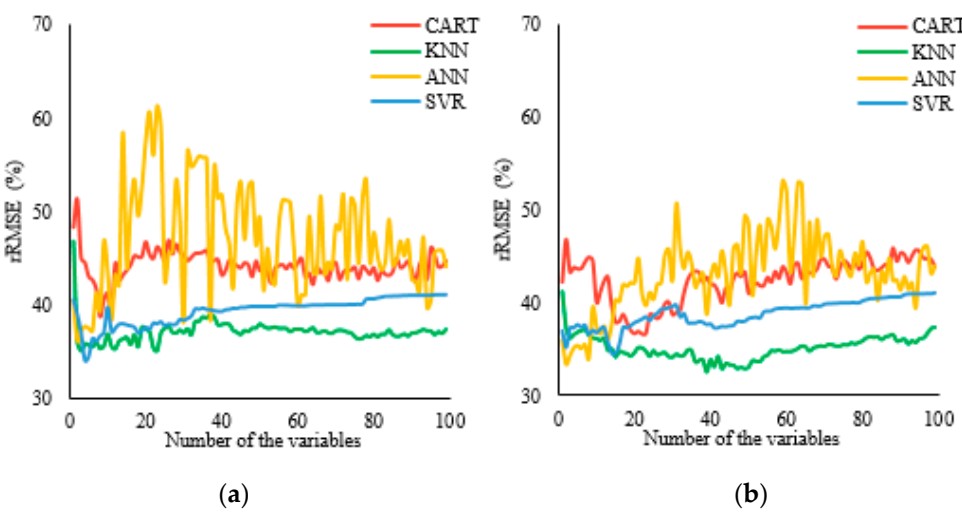

**Figure 4.** The rRMSE of estimated GSV varied with the number of combined variables. (**a**) RF; (**b**) PCC.

To construct the proposed variable selection criterion and establish the optimally combined variables, four variable selection methods (RF, DC, MIC, and PCC) and four machine learning models (CART, KNN, SVR, and ANN) were employed. Additionally, the proposed strategy was challenged against the FORW method. The results of the combined variables are listed in Table 4. The first selected variable per approach depended on the ranking method, while the numbers of operations and the selected variables were related to the employed models for estimating GSV. Using the suggested variable selection criteria with various single ranking methods, the selected variables ranged from 4 to 9. By using the FORW, the corresponding number varied from 2 to 10 and the number of operations from 295 to 1034.

**Table 4.** The results of the proposed criterion for variable selection.

| Variable Selection Criterion | Method of Ranking | Models | The First Selected Variable | Number of Variables | Number of Operations |
|---|---|---|---|---|---|
| Forward | FORW | CART | band4 | 2 | 295 |
|  | FORW | KNN | band4 | 10 | 1034 |
|  | FORW | ANN | EVI_2 | 4 | 486 |
|  | FORW | SVR | EVI | 6 | 672 |
| The proposed criterion for variable selection | RF | CART | EVI_2 | 6 | 41 |
|  | RF | KNN | EVI_2 | 6 | 27 |
|  | RF | ANN | EVI_2 | 4 | 51 |
|  | RF | SVR | EVI_2 | 6 | 43 |
|  | DC | CART | band4_M | 4 | 67 |
|  | DC | KNN | band4_M | 5 | 26 |
|  | DC | ANN | band4_M | 4 | 58 |
|  | DC | SVR | band4_M | 7 | 64 |
|  | MIC | CART | band4_M | 4 | 95 |
|  | MIC | KNN | band4_M | 9 | 37 |
|  | MIC | ANN | band4_M | 5 | 46 |
|  | MIC | SVR | band4_M | 5 | 44 |
|  | PCC | CART | RVI | 7 | 20 |
|  | PCC | KNN | RVI | 5 | 21 |
|  | PCC | ANN | RVI | 4 | 65 |
|  | PCC | SVR | RVI | 7 | 59 |

The proposed variable selection criteria show great potential for promoting efficiency. Indeed, the number of operations required by our strategy ranged from 20 to 95, significantly less than FORW (Table 4), with Figure 5 illustrating the variable selection processing utilizing various approaches. By employing FORW's criterion, the number of operations significantly increased (Figure 5a,b). Filtering the alternative variables through the collinearity test increased the efficiency of determining the number of optimally combined variables for all four ranking methods (Figure 5c–f).

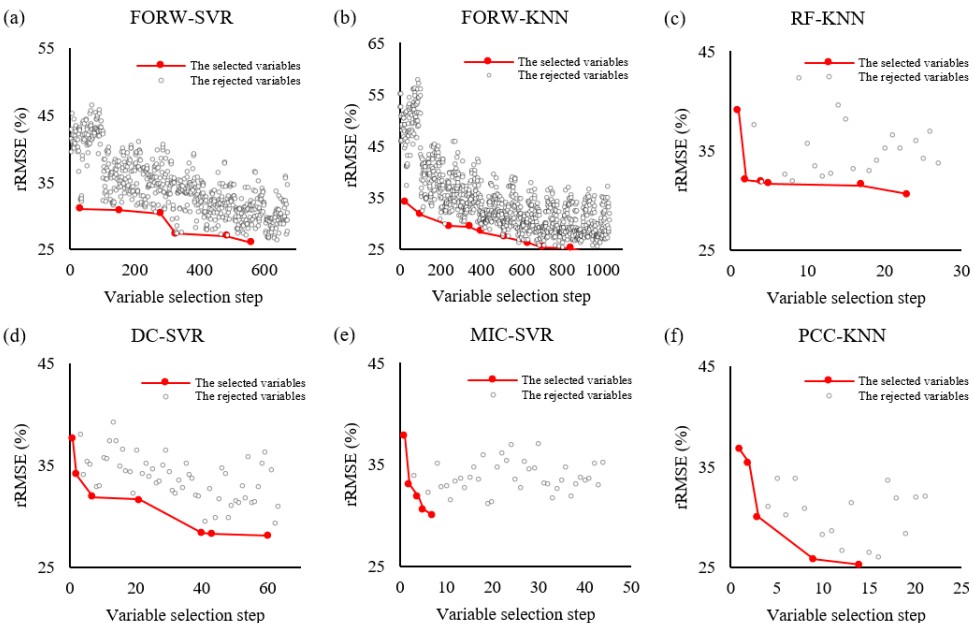

**Figure 5.** The processing of variable selection with various approaches: (**a**) FORW-SVR, (**b**) FORW-KNN, (**c**) RF-KNN, (**d**) DC-SVR, (**e**) MIC-SVR, (**f**) PCC-KNN.

Furthermore, the accuracy of estimating the forest GSV is also a critical factor in evaluating the proposed variable selection criterion. Figure 6 presents the rRMSE and the $R^2$ histogram extracted from different variable selection approaches. The values of rRMSE ranged from 33.32% to 38.20% for FORW and from 30.40% to 37.75% for the proposed criterion utilizing four methods (RF, DC, MIC, and PCC), respectively. Considering the FORW's $R^2$ values, these were slightly lower than those of the proposed criterion. Therefore, the proposed criterion efficiently obtained the optimally combined variables without decreasing the accuracy of mapping forest GSV.

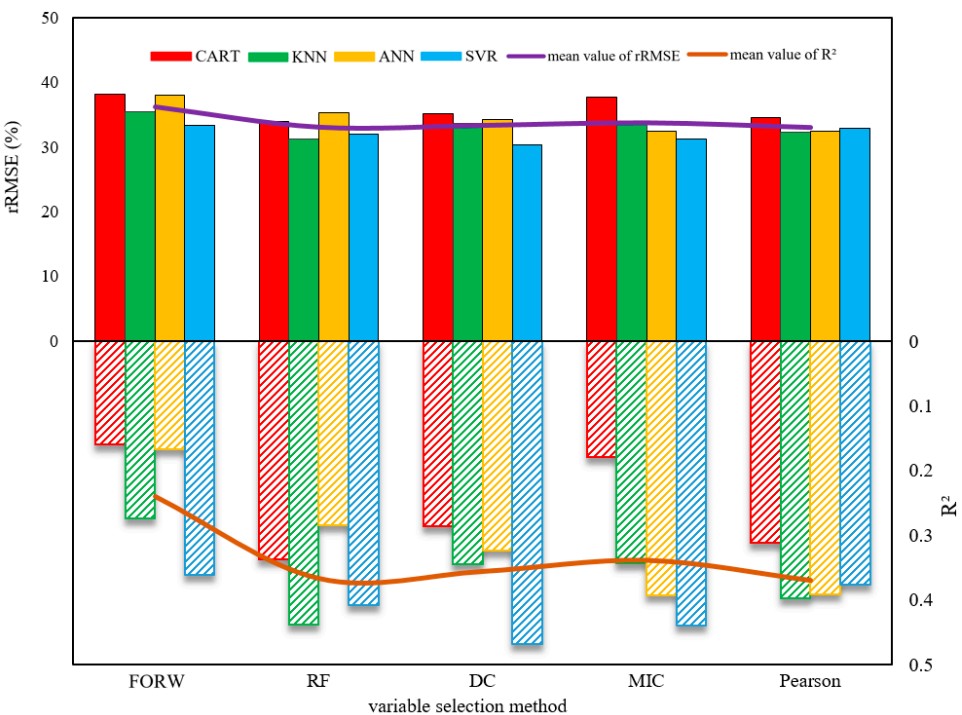

**Figure 6.** The accuracy indexes of forest GSV for FORW and proposed criterion with various ranking methods.

### 3.2. The Result of the Secondary Ensemble

As already noted, all ground measured samples were divided into three sets: training (36 samples), validation (18 samples), and testing (27 samples). For each base learner model and ranking method, the approaches of Bagging and AdaBoost were applied to reduce the uncertainty of constructing the model based on the selected training and validation set. Table 5 and Figure 7 illustrate the results of the first ensemble of forest GSV.

**Table 5.** The results of the first and secondary ensemble.

| Ranking Methods | First Ensemble (Bagging and AdaBoost) | | | Secondary Ensemble (IWA) | | | |
| | Number of Variables | rRMSE (%) | $R^2$ | Number of Models | Number of Related Variables | rRMSE (%) | $R^2$ |
|---|---|---|---|---|---|---|---|
| RF | 4~6 | 21.91~30.28 | 0.47~0.72 | 3 | 7 | 20.14 | 0.77 |
| DC | 4~7 | 23.41~28.89 | 0.52~0.68 | 8 | 14 | 21.34 | 0.74 |
| MIC | 4~9 | 23.60~31.49 | 0.43~0.68 | 5 | 13 | 19.89 | 0.77 |
| PCC | 4~7 | 21.93~28.83 | 0.52~0.72 | 8 | 15 | 18.89 | 0.79 |

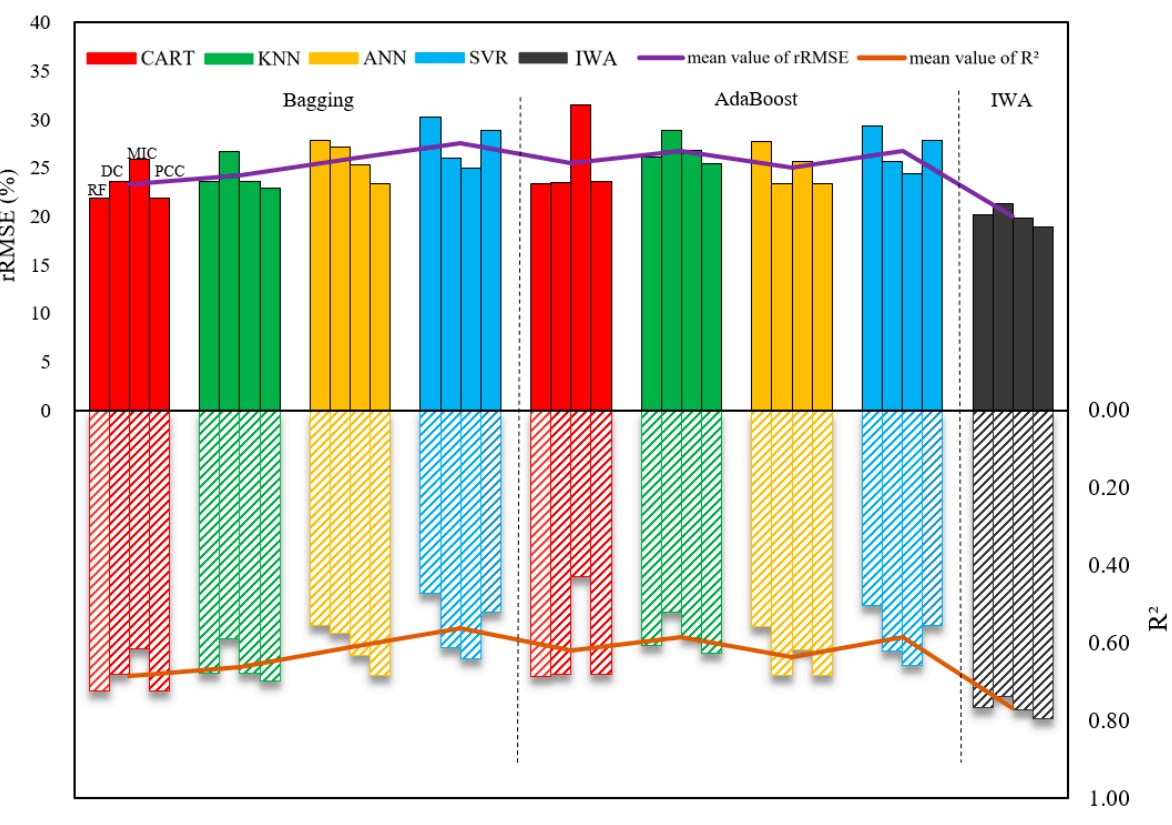

**Figure 7.** Forest GSV accuracy indices for the first and secondary ensemble.

Considering the results from four base learners (Figure 6), the rRMSE values exceeded 30%, and the $R^2$ was less than 0.5. After the first ensemble exploiting Bagging and AdaBoost (Table 5), the accuracy of results was obviously improved, and the rRMSE and $R^2$ values ranged from 21.93% to 31.49% and from 0.43 to 0.72, respectively. Furthermore, the range variance of the precision index was relatively small for Bagging and AdaBoost. However, the accuracy of estimating GSV is still highly related to the employed models.

We reduced the performance variance when various models and combined variable sets were applied by involving a secondary ensemble that integrated the results from different base learners. This study regarded as base learners four machine learning models (CART, KNN, SVR, and ANN) and employed the IWA to construct the model of the secondary ensemble. For each ranking method, eight results were extracted from the four base learners through Bagging and AdaBoost. The first ensemble results were joined to the secondary ensemble model one by one, and the contribution to decreasing the rRMSE values was regarded as a threshold to determine whether the model should be removed. Finally, after exploiting the secondary ensemble, the corresponding results from four ranking methods are presented in Table 5 and Figure 7.

Unlike the first ensemble models, the secondary ensemble models have the advantages of base learners, and the estimated forest GSV is more reliable than the results from the first ensemble. Table 5 illustrates that the rRMSE values from the secondary ensemble models (ranged from 18.89% to 21.34%) were less than those from the first ensemble models (ranged from 21.93% to 31.49%), and the variance of the accuracy was narrowed to within 3% (Figure 7). It is found that the secondary ensemble decreased the variance between base models and ranking methods.

To further analyze the results from various approaches, the scatterplots between predicted and measured GSV with various models are shown in Figure 8. For the base learners without integration (Figure 8a,b), the accuracies of the estimated forest GSV were highly related to the models and the variable selection methods, and overestimated results

often appeared. The errors between the predicted and measured GSV were reduced using the first ensemble (Figure 8c,d). After the secondary ensemble, the variance between the base models decreased, especially for the samples with high GSV. The experimental results proved that the ensemble between various base learners had improved the mapping forest GSV accuracy.

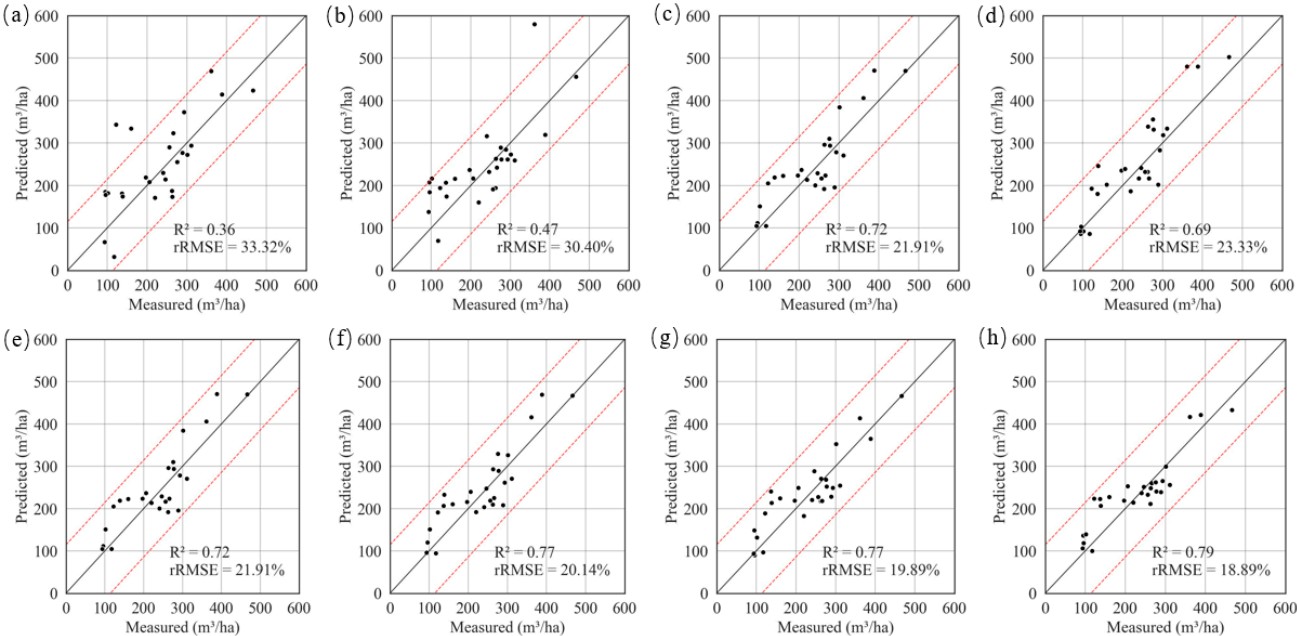

**Figure 8.** Scatter plots between predicted and measured GSV from various approaches: (**a**) FORW-SVR, (**b**) DC-SVR, (**c**) RF-CART-Bagging, (**d**) RF-CART-AdaBoost, (**e**) RF-IWA, (**f**) DC-IWA, (**g**) MIC-IWA, and (**h**) PCC-IWA.

### 3.3. Mapping the Forest GSV

To map the forest GSV, the results of the secondary ensemble utilizing four ranking methods were derived from the first ensemble models through the IWA (Figure 9). The latter figure illustrates that the estimated GSV ranged from 50 $m^3$/ha to 250 $m^3$/ha. To further analyze the capability of the ensemble, the variance between the ensemble models using various ranking methods was extracted from the first and secondary ensemble models, and the histogram of the variance between the ensemble models is shown in Figure 10. Five groups of variance GSV from the first ensemble (Bagging and AdaBoost) and secondary ensemble (IWA) were extracted from the mapped forest GSV. For the results of the secondary ensemble, the mapped GSV extracted from the IWA with PCC was regarded as a reference, and the variance of GSV between PCC and other methods was calculated. Regarding the results obtained from Bagging and AdaBoost, the mapped GSV utilizing the CART model with four ranking methods were employed, and the results extracted from the first ensemble with RF were regarded as a reference. Additionally, the Bagging and AdaBoost results with the variable ranking of PCC and the mapped GSV extracted from the first ensemble (Bagging and AdaBoost) with CART were also regarded as a reference.

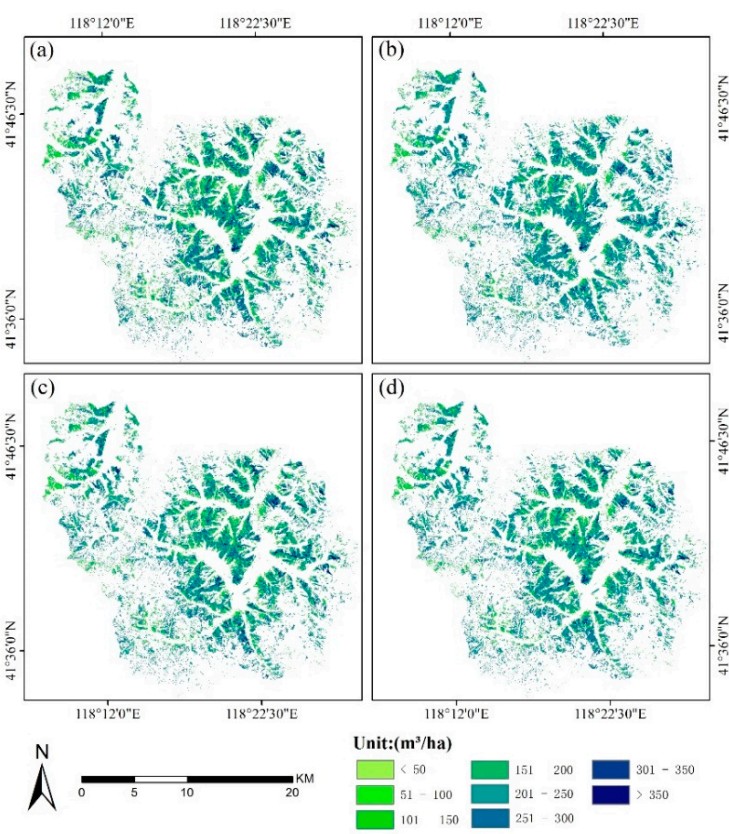

**Figure 9.** The map of planted coniferous forest in the study area: (**a**) RF-IWA, (**b**) DC-IWA, (**c**) MIC-IWA, and (**d**) PCC-IWA.

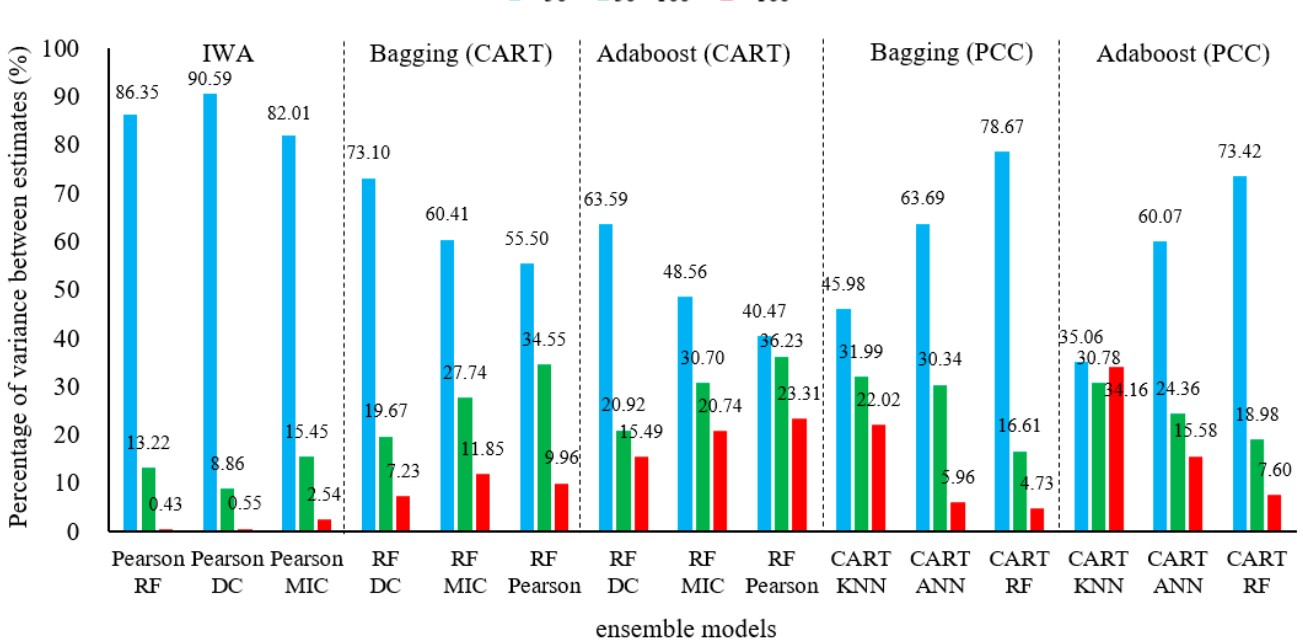

**Figure 10.** The histogram of the variance between various ensemble models.

The variance between the two mapped GSV was divided into three parts: less than 50 m$^3$/ha, 50 to 100 m$^3$/ha, and greater than 100 m$^3$/ha. For the same ranking method variables (PCC), the estimated GSV varied with the employed base model and ensemble algorithms, and the percentage of statistic variance GSV (greater than 100 m$^3$/ha) was up

to 34.16%. For the same base learner with different ensemble algorithms, the mapped GSV uncertainty still hindered obtaining reliable forest GSV because of the different capabilities of the various ensemble.

After applying the secondary ensemble, the percentage of statistic variance GSV (less than 50 m$^3$/ha) ranged from 82.01% to 90.59%, and the percentage from 0.43% to 2.54% for variance exceeding 100 m$^3$/ha. The mapped GSV uncertainty was significantly reduced regardless of the four variables ranking methods. Therefore, it is proved that the secondary ensemble exploiting the IWA afforded a reliably mapped forest GSV by reducing the variance between the selected models.

## 4. Discussion

### 4.1. Variable Selection

The accuracy of estimating GSV is strongly related to the variable selection methods [8,42–44]. However, it is still controversial to determine the combination of the optimal variable when mapping the forest GSV employing imagery from various sensors. It is well known that the standard variable selection methods consider only the relationship between variables and GSV through a quantitative evaluation process, such as the PCC correlation coefficient. In previous studies, these approaches were often used to assess the sensitivity and set thresholds directly [16,45–48]. However, the GSV estimation accuracy depending on the thresholds was not stable (Figure 4). The FORW method combined variable selection methods with the estimated models, with its higher accuracy obtained at the expense of computational complexity [49,50].

Our study, in the proposed variable selection criterion, considered the variables' combination effect and collinearity. Based on our experimental results, the number of operations required by the proposed approach was significantly less (ranged from 21 to 95) than these of FORW (ranged from 295 to 1034), and the time for searching the optimal combination of variables was significantly reduced. Furthermore, the estimated GSV accuracy using the proposed variables selection method was slightly higher than the FORW. Therefore, the proposed variable selection criteria can efficiently obtain the optimally combined variables without decreasing the accuracy of GSV estimation (Table 4 and Figure 5).

### 4.2. Ensemble Model

After selecting the optimal variables combination, the accuracy and reliability of the estimated GSV are highly related to the capability of the employed models. Generally, machine learning approaches have been widely used to solve complex and multi-variable problems [51–53]. These approaches were also employed to map forest GSV to combine the advantages of multi-source remote sensing data. Indeed, the machine learning methods achieved higher estimated GSV accuracy compared to parameter models. However, the estimated GSV uncertainty was induced when a single machine learning model was selected [33–35]. Similar results were also obtained in our study, with the rRMSE and R$^2$ values obtained by a single machine learning model ranging from 30.40% to 37.75% and from 0.18 to 0.47, respectively. Thus, the mapped GSV uncertainty was generated by the capability of the employed models [7,34,35].

To reduce the impact of uncertainty, various ensemble algorithms were employed, combining several base models. The ensemble algorithms had a better generalization capability and presented a more robust performance than the single model. In this study, the estimated GSV accuracy greatly improved with the rRMSE values ranging from 21.91% to 31.49% after the first ensemble with Bagging or AdaBoost. Figure 11 illustrates the relationship between the residuals and ground measured GSV. The residuals derived from the first ensemble (Figure 11c,d) were smaller than those derived from the single machine learning model (Figure 11a,b).

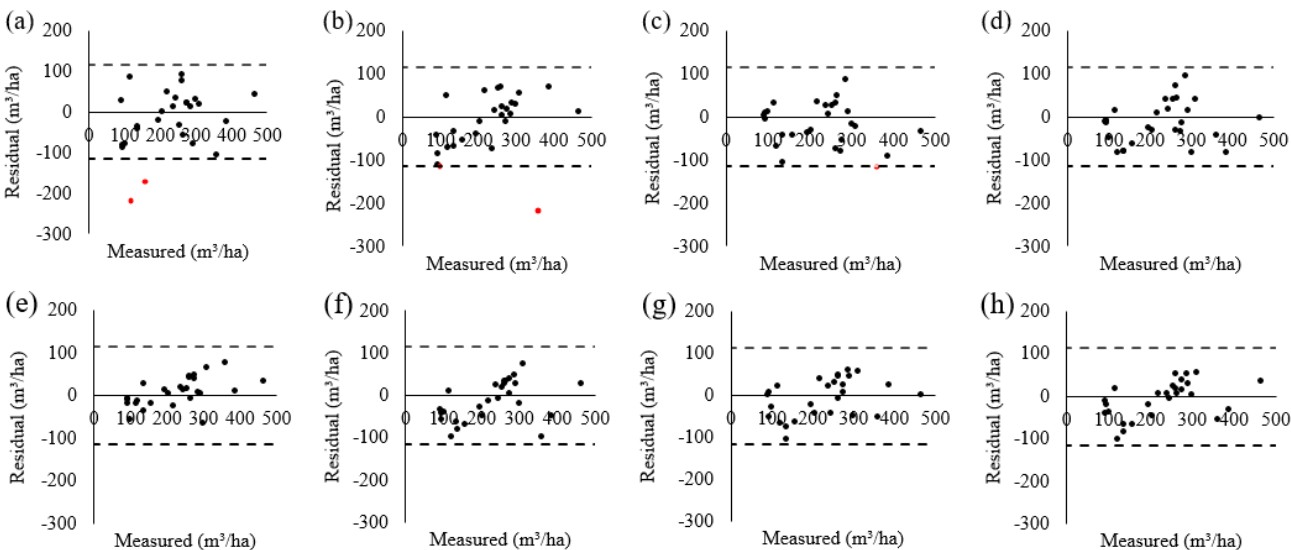

**Figure 11.** The plots of relationship between the residual and ground measured GSV; (**a**) is FORW-SVR; (**b**) is DC-SVR; (**c**) is RF-CART(Bagging); (**d**) is RF-CART(AdaBoost); (**e**) is RF(IWA); (**f**) is DC(IWA); (**g**) is MIC(IWA); (**h**) is PCC(IWA). The red dots indicate that its residual exceeds 50% of the average GSV in all samples.

To reduce the various model uncertainty, we proposed a secondary ensemble to obtain the merits of each selected base model. The methods involved in the first ensemble reduce the model uncertainty caused by the samples, while the uncertainty from the different models themselves is omitted. Exploiting the secondary ensemble decreased the variance between various models, and the proposed IWA further enhanced the forest GSV reliability. The percentage of statistic variance GSV (smaller than 50 m$^3$/ha) ranged from 82.01% to 90.59% for the IWA. The results of existing studies exploiting ensemble algorithms for learners with different [7,34] or the same base [19,54,55] learners have also shown an appealing estimation accuracy. Therefore, it is confirmed that the proposed combination strategy of variable selection and ensemble algorithm has a great capability to obtain reliable forest GSV.

The distribution of GSV in this study area was also mapped in previous studies [7,56]. In this paper, we obtained a similar GSV estimation accuracy, but the remote sensing datasets used, the variable selection methods and the ensemble algorithms were different, and our objectives were not the same. Multiple variable selection methods and ensemble algorithms were compared and used in this study to reduce uncertainty and obtain more accurate and stable estimation results.

## 5. Conclusions

This study proposed a combined strategy of improved variable selection and ensemble algorithm to map the reliable GSV of planted coniferous forests. Considering the variables' combination effects and collinearity, an improved variable selection criterion was initially applied to efficiently select the combination of the optimal variable extracted from Sentinel-1A and Sentinel-2A datasets. Four machine learners were regarded as base learners, and the combined strategy was constructed to reduce the uncertainty of the mapped GSV caused by selecting samples and models. It is proved that the proposed variable selection criterion could efficiently obtain the optimal combination of variables without reducing the accuracy of mapping forest GSV. The results also confirmed that the first ensemble models using Bagging and AdaBoost were more stable and accurate than a single machine learner.

Furthermore, the uncertainty of various models is significantly decreased by utilizing the secondary ensemble with the IWA. The percentage of statistic variance GSV (less than 50 m$^3$/ha) ranged from 82.01% to 90.59% for the IWA. It is proved that the proposed combination strategy has great potential to reduce the variance of estimated GSV between various variable selection approaches and employed base models. However, the

methodology of our work was subject to the ground measured samples and various images. Therefore, further study will be conducted to prove the feasibility of our strategy in other complex forests. The datasets and codes used in this study have been uploaded to Zenodo 10.5281/zenodo.5641318, enabling other researchers to improve data processing methods.

**Author Contributions:** Conceptualization, X.X., J.L. and H.L.; methodology, X.X., Z.L. and J.L.; software, X.X.; validation, X.X., J.L. and H.L.; formal analysis, Z.Y. and X.L.; investigation, X.X., Z.L., J.L., H.L. and Z.Y.; resources, X.X., Z.L., J.L. and Z.Y.; data processing, X.X. and Z.L.; original draft, X.X.; review and revision, X.X., J.L. and H.L.; final editing: H.L.; visualization, X.X., J.L. and H.L.; supervision, H.L. and J.L.; project administration, X.X. and J.L.; funding acquisition, H.L. All authors have read and agreed to the published version of the manuscript.

**Funding:** This research was funded by the National Key R&D Program of China project "Research of Key Technologies for Monitoring Forest Plantation Resources" (2017YFD0600900) and Innovative province and Construction special funds of Hunan Province "Intelligent measurement and monitoring technology of forest stock, biomass and carbon storage based on multi-source data of land, space and sky" (2020NK2051).

**Data Availability Statement:** The observed GSV data from the sample plots and spatial distribution data of forest resources presented in this study are available on request from the corresponding author. Those data are not publicly available due to privacy and confidentiality. The Sentinel-1A (level-1 GRD) product and the Sentinel-2 (level-1C) product were obtained from the Copernicus data center website at https://scihub.copernicus.eu/ (accessed on 5 October 2019).

**Conflicts of Interest:** The authors declare no conflict of interest.

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
