# Peer review of "A Combined Strategy of Improved Variable Selection and Ensemble Algorithm to Map the Growing Stem Volume of Planted Coniferous Forest"

_remotesensing, doi:10.3390/rs13224631_

Round 1

Reviewer 1 Report

I've only reviewed half of the paper due to the lack of reproducibility (no data and code attached) in combination with the high amount of minor and major comments listed below (which only emerged from the first half of the manuscript). I'd like to ask the authors to make the data available to support reproducibility or to provide strong arguments why this is not possible or why they haven't at least provided the code and Sentinel data for public access.

In addition, authors are encouraged to do a spell and grammar check before submission and also check for whitespace issues.

I am happy to review the manuscript again if these two points are fulfilled in the next round.

## Major comments

- The abstract talks about "proposed variable selection criterion" but this criterion is not introduced.
- The introduction of filters, wrapper and embedded feat methods (L62 - 74) is a bit mixed and squashed and should be rephrased.
- The term "estimation models" does not exist and should be changed.
- Language and grammar, especially the use of adjectives and adverbs, needs to be revised.
- The use of "quickly" seems to be wrong across the whole manuscript and should probably be avoided.
- Figure2: Text is too small and not readable.
- The authors should explain why they chose the indices listed in Table 3.
- There are quite some whitespace issues (too many, no whitespace after a period) and it seems to me no spell check was applied before the submission.
- L221: training (36 samples), validation (18 samples), and testing (27 samples).

    Validation and test sets are the same usually - it is unclear why there is a difference being made and how this affects the performance estimation conducted.

- The observed GSV data from the sample plots and spatial distribution data of forest resources presented in this study are available on request from the corresponding author. Those data are not publicly available due to privacy and confidentiality.

    It is unclear why the GSV data can not be made available for the given reasons as there are usually no privacy concerns to GSV data. In addition, the used Sentinel images should be made available in a research compendium or online data storage for reproducibility. The same applies to the code used for model fitting and evaluation.

## Minor Comments

- L21: with standard methods of variable importance ranking, e.g., RF, DC, MIC, and Pearson

    Could you please explain what are "standard methods of variable importance ranking"? Acronyms were not introduced yet at this point.

- L22: with an improved voting regression (Voting)

    If (Voting) should serve as an acronym for "voting regression", this should be made more clear by rephrasing this.

- L29: the differences of GSV mapped by the second integration

    I think "differences" should be replaced by "variance" here.

- L54: However, the cost of acquiring LiDAR data limits its largescale application.

    This sentence is not valid English, please rephrase.

- L62: Given the various alternative variable types, variable selection methods are the critical step to obtain the optimally combined variables for forest GSV estimation, with the accuracy of the results highly depending on the sensitivity of the selected variables [6,7,14].

    Could you please clarify what are "alternative variable types"? Variable selection (usually called feature selection) is not important because of alternative variable types (which means numeric, nominal or character type variables) - in fact it is completely unrelated to the variable type.

- L66: The filters are based entirely on a single criterion, such as random forest (RF), distance correlation coefficient (DC), maximal information coefficient (MIC), and Pearson correlation coefficient [7,16–18].

    To my knowledge RF is not a filter method - it has an integrated variable selection and is an algorithm for model fitting. Please clarify.

- L75: Hence, it is essential to obtain the optimal variable combination for mapping the forest GSV quickly.

    Unclear what that means - especially the term "quickly" in this context.

- L79: With the increasing of remote sensing images

    With the increase of [...]

- L86: integration learning approaches, e.g., RF [31], Bagging [34], and AdaBoost [35],

    What are "integration learning approaches"?

- L163: Extraction Variables

    Should probably read "Extraction of Variables" or "Variable Extraction"

- L183: As we know [...]

    I suggest to avoid phrasings like this one.

Author Response

Please see the attachment for comment responses, and the corresponding revised manuscript has been uploaded.

Reviewer 2 Report

In this paper the authors proposed a combined variable selection and integration strategy for mapping forest GSV that afforded quickly obtaining the optimally combined variables and reduced the model uncertainty. The GSV estimation accuracy utilizing remote sensing imagery is highly related to the variable selection methods and estimation models. To reduce the uncertainty of mapping GSV, selecting the appropriate remote sensing images, variables, and models depending on the various forest and external environments is mandatory. The authors use the data and methods available in the literature to perform this work. The results achieved are reasonable and reduced the GSV mapping uncertainty when compared with current variable selection approaches and estimation models. The paper is clearly presented.

Some corrections and suggestions

L.105- Section 3 presents the experimental results, and finally, Section 4 concludes this work.  ---  Section 3 presents the experimental results, Section 4 presents the discussion, and finally, Section 4 concludes this work.

L.110- of 25,958 hm2),  ---   of 25,958 km2),

L.122- of This Research  ---  of this research

L.151- The band characteristics and polarizations are listed in Table 2, while the variables were mainly derived from the spectral bands of Sentinel-1A and the backscattering coefficient of VV and VH polarizations with a matched spatial resolution.   ---  This statement needs to be revised to be clear for the readers.

L.178 – Table 3 – I suggest to include the references for the vegetation indexes used.

L.279-sensitivity of and set thresholds directly  ---    sensitivity and set thresholds directly

L.289- Therefore, the proposed criterion involving the collinearity test quickly Geobtained the optimally combined variables without decreasing the mapping forest GSV accuracy ---  This statement needs to be revised to be clear for the readers.

Author Response

(The authors gave the same response as above.)

Round 2

Reviewer 1 Report

Thanks for answering in such great detail, I acknowledge the authors motivation.

Datasets & reproducibility

Thanks for uploading your data and code. I would have been great to have some kind of README which states how to use the scripts in combination with the data and in which order.

Uploading such data as suppl. material is one way, I suggest to look into dedicated data upload providers such Zenodo next time. Especially when data becomes big, suppl. material becomes problematic whereas the mentioned providers support data uploads of up to 50 GB.

Last, I'd like to request to mention in the manuscript that data and code are available in the suppl. material (or elsewhere in case the authors decide to upload to a dedicated hoster).

Validation vs test sets

Rather than using the train-test-validation concept, I suggest to look into nested cross-validation next time, which includes tuning within a cross-validation and avoids having to three-way split the dataset upfront, thus leaving more samples for each of the conducted stages.

Language

Even though an official language review was conducted according to the authors, I see very few substantial changes, especially grammar changes. I am usually quite sceptical when it comes to such certificates, but I am not the one to judge. 

Summary

Given that the authors tried to implement most of the requested changes, I'll recommend for publication and hope that the data/code will be mentioned in the manuscript and also included in further submissions right from the start :)

Author Response

Thank you for your comments, and please see the attachment.
